# Exploring the barriers and facilitators to the uptake of smoking cessation services for people in treatment or recovery from problematic drug or alcohol use: A qualitative systematic review

**Eric Osayemwenre Iyahen**[1]*, **Osazee Omoregbe Omoruyi**[2], **Neneh Rowa-Dewar**[3], **Fiona Dobbie**[3]

**1** Department of Public Health, Ministry of Health, Benin City, Edo State, Nigeria, **2** Ministry of Health, Benin City, Edo State, Nigeria, **3** Usher Institute, University of Edinburgh, Edinburgh, United Kingdom

* ericiyahen@gmail.com

## Abstract

### Background

Smoking prevalence and the associated poor health and mortality is significantly higher among people with/recovering from problematic drug or alcohol (PDA) use in comparison with the general population. Evidence from existing systematic reviews shows smoking cessation enhances rather than compromises long-term abstinence from alcohol or drug use. However, these systematic reviews lack important contextual detail around the reasons why uptake of, and successful engagement with existing stop smoking services remains low for people in treatment or recovery from PDA use. This systematic review explores qualitative data on the barriers and facilitators to the uptake of smoking cessation services for people in treatment or recovery from PDA use. This key objective addresses the limited inclusion of qualitative studies in previous systematic reviews on this issue.

### Methods

A qualitative systematic review was conducted with searches across four electronic databases (PubMed, Embase, PsycINFO and Cumulative Index to Nursing and Allied Health Literature [CINAHL]). All studies that had a qualitative component about free smoking cessation/reduction programmes for people in treatment or recovery from PDA use were included. Studies that examined electronic smoking or services that required a fee were excluded. Study quality was assessed using National Institute for Health and Care Excellence checklist. Qualitative synthesis involved inductive thematic analysis. (PROSPERO Registration number: CRD42022298521).

**Data Availability Statement:** All relevant data are within the manuscript and its Supporting Information files.

**Funding:** The author(s) received no specific funding for this work.

**Competing interests:** The authors have declared that no competing interests exist.

## Results

8809 potentially eligible articles were identified, 72 full texts were screened and ten articles were included for full review. Barriers to uptake and engagement with existing stop smoking services centered on three key themes: perception of public health importance, programme structure, and intervention elements. Facilitators included supportive treatment environment and optimization of support/staff resources for smoking cessation service delivery.

## Conclusion

Recommendations included influencing a change in the way people perceive the importance of smoking cessation activities during PDA use treatment or recovery. There was also some emphasis on the need to create the right environment for sustained adherence to treatment or recovery plans, and deliver the interventions within the health system as comprehensive care. The limited qualitative evidence on community-based and outpatient services highlights a research gap.

## Introduction

Tobacco smoking prevalence among people with/recovering from problematic drug or alcohol (PDA) use is significantly higher than in the general population. The global smoking rates for this group and the associated illnesses and deaths have been estimated to be 2–4 times higher than in the general population [1–6]. People who have PDA use are more likely to suffer from complex comorbidities (including respiratory ill-health associated with smoking) and mental health challenges. Studies have also shown that people with PDA use are more likely to die from smoking than from their alcohol or drug use [7–10].

There are different forms of smoking cessation support available to the general population in different countries [11, 12]. This support involves the use of pharmacological and/or behavioural approaches [5, 13–15]. Pharmacological strategies involve the use of medications such as nicotine replacement therapy (NRT), varenicline and bupropion, while behavioural approach involves individual or group counseling sessions [14, 16–20].

There is a significant evidence base looking at the effectiveness of pharmacological and/or behavioural smoking cessation approaches for the general population. Some studies found some evidence that NRT increases smoking abstinence when used either alone or in combination with behavioural counselling sessions [4–6, 13–15, 21, 22]. Despite this, the uptake of, and successful engagement with existing stop smoking services has remained generally low for the general population especially in the long-term [22–26]. In comparison, the rate at which people in treatment/recovery from PDA use utilize and successfully engage with existing stop smoking services is consistently lower than in the general population [5, 6, 27–29]. Guidance for alternative approaches which promote smoking harm reduction for people in PDA use treatment/recovery have been recently provided by the National Institute for Health and Care Excellence (NICE) [30]. This harm reduction strategy acknowledges that problematic substance use is linked to the socio-economic conditions under which the individual lives [31, 32].

Existing systematic reviews highlight that stop smoking services are effective and also increase the odds of long-term abstinence for people in treatment/recovery from PDA use [4–6]. However, these systematic reviews only considered Randomized Controlled Trials (RCTs) focusing on the effectiveness of existing interventions not the barriers and facilitators

to uptake and successful engagement with the services. There is therefore, a lack of contextual detail around the reasons why uptake of, and successful engagement with existing stop smoking services remains low for people in treatment or recovery from PDA use. Addressing this gap is the purpose of this qualitative systematic review. The overall aim of the review was to examine the barriers and facilitators to engagement with smoking cessation programmes for people with/recovering from PDA use.

Smoking cessation programmes were defined as specialized smoking services offered to people who want to stop smoking tobacco [12, 33]. Smoking cessation services are usually delivered within the health systems and evidence to evaluate the delivery and effectiveness exists in developed countries such as the United Kingdom (UK), United States of America (USA) and Australia [34–37]. People enrolled in either residential/inpatient or outpatient treatment programmes for PDA users were considered to be 'in treatment', while those who had already received some form of support for their PDA use were considered as people 'in recovery'. The delivery of this systematic review was guided by two review questions:

1. What barriers do people with PDA use face when seeking help to cut down or stop smoking?

2. What factors promote the successful engagement and positive experiences of interaction with stop smoking services for people with/recovering from PDA use?

## Methods

### Study design

This systematic review followed the Preferred Reporting Items for Systematic Reviews and Meta-Analyses (PRISMA) [38] and Enhancing transparency in reporting the synthesis of qualitative research (ENTREQ) [39] guidelines. A protocol for this review was registered with International Prospective Register of Systematic Reviews (PROSPERO) [Identifier: CRD42022298521], S1 Appendix. This review was approved by the Usher Masters Research Ethics Group (UMREG) at the University of Edinburgh, Scotland. Supplementary information for this systematic review includes the PRISMA checklist (S1 File) and ENTREQ checklist (S2 File).

### Databases and search strategy

The search strategy was defined using the Participants, Interventions, Comparisons and Outcomes (PICO) framework to which Setting (S) and Type of study (T) have been added [40]. PubMed, Embase, PsycINFO and CINAHL databases were electronically searched for journal articles published from each database's inception to 19th November 2022. The searches were done using a combination of free text keywords and Medical Subject Headings (MeSH) terms from each database (S1 Fig). Key words were associated with smoking, smoking cessation, substance abuse and treatment/recovery. The search strategy was first developed and tested in PubMed by one reviewer (EI). This strategy was then applied to each of the other databases between 15th and 18th of November 2022. Finalized searches were run across all four databases on 19th November 2022 (S1 Fig). Manual searches of reference lists of key articles were done for supplementary references.

### Inclusion and exclusion criteria

In contrast to existing systematic reviews of relevance to this topic which only include randomized control trials (RCTs), this review focused on qualitative studies, mixed method

studies (qualitative component only) or qualitative components in the process evaluation of RCTs. Journal articles published in English from database inception to the search date were deemed eligible for review using pre-specified search terms. Primary research studies reporting qualitative findings from all countries where free smoking cessation programmes have been implemented were included. Qualitative studies that reported on smoking cessation support or services that required a fee were excluded. Studies that examined electronic smoking for smoking cessation were also excluded. Systematic reviews were not included but reference lists were scanned as sources of primary studies. People with PDA use were also categorized as having drug dependence or substance use issues [41–43].

## Study selection and data extraction

The finalized search terms were run through the four databases and references were exported to Covidence where duplicates were removed and references were stored. Covidence is an online systematic review programme with a simple, user-friendly interface which allows multiple researchers to work in an efficient manner on the same project, through the steps of the systematic review process [44, 45]. Titles of search results were assessed to remove references which were not related to the search parameters. This was done by the principal reviewer (EI) according to pre-determined inclusion criteria, with 10% of the studies excluded (randomly selected) double-checked by a second reviewer (OO) independently. This was aimed at enhancing objectivity and robustness in ascertaining their relevance. Titles and abstracts of studies that were potentially eligible for inclusion underwent duplicate independent review by the two reviewers, based on the same eligibility criteria. Full texts of studies that met our inclusion criteria were retrieved and independently assessed for eligibility by two authors (EI) and (OO). In all cases, disagreements were resolved by discussion with a third reviewer (FD) and a final list of studies for full review was generated.

Data from included articles were extracted using a customized summary sheet as template. The template was first tested with three articles independently by two reviewers and then used to record study aims and country where research was conducted. Other data items included sample characteristics, type of support received and where it was offered, as well as findings and conclusions. Summary of the findings were those of relevance to the objectives of this systematic review only.

## Study quality assessment

The principal author (EI) independently assessed the quality of 70% of the included studies using the NICE critical appraisal checklist designed for qualitative studies [46, 47] (S3 File). This was achieved using the guidelines provided for the effective use of the checklist [46] (S2 Appendix). The tool assessed 14 items under unique domains of the qualitative research methods for each included qualitative study, to test the validity of the theoretical approach. An overall assessment of the included studies was done and one of three final gradings which are ++, + and−was given. This grading was based on the number of the checklist criteria that have been fulfilled, and if not fulfilled or adequately described, whether the conclusions were likely to alter or not. This process followed a double blinded assessment strategy where another reviewer (OO) independently assessed 30% of the included studies.

## Data synthesis

A thematic analysis of study findings was conducted using an inductive approach [48]. An inductive thematic analysis ensures that the meanings are drawn directly from the data and not shaped by a pre-conceived idea or theoretical framework [48, 49]. Two authors (EI) and

(OO) first read through the quotes in the text twice to bring about familiarity with the qualitative data in the included studies. A codebook was developed where data of potential relevance to the research questions were highlighted. Initial codes representing participants' direct opinions were written verbatim. The initial codes were generated from each data item in the entire body of data. This was done in sequence to ensure inclusivity. Codes were re-read by (OO) to maintain consistency. The initial codes were organized into second-level descriptive codes and codes that conveyed similar meanings were grouped together. The coding process identified the views of service users and treatment providers which were analyzed together. Participants' verbatim quotes that expressed very similar ideas were first aggregated into third level analytical sub-themes. All closely-related sub-themes that intersected and were precursor to one another but conveyed a central message were organized into themes. The emerging themes were those that directly addressed the research questions.

## Results

### Search results

The electronic database searches identified 8809 studies which were merged across Covidence. After duplicates were removed, 8529 studies passed through the screening stage, 420 database records underwent title and abstract screening, of which 72 studies met the eligibility for full text review. After screening, seven articles met the inclusion criteria and were eligible for qualitative synthesis. Three more studies were manually sourced by scanning the reference lists of the included studies and added during the inclusion phase of the study selection process. This made the total number of studies for the qualitative systematic review ten (see Fig 1 for PRISMA flow diagram). Seven of the included studies were published between 2013 and 2020; three were published before the year 2010. The reviewed studies were all conducted in high income countries, with eight of the studies carried out in the USA [50–57]. Of the remaining two studies, one was conducted in the UK [58] and another in Australia [59].

### Study characteristics

The included studies all reported qualitative data on tobacco use and/or cessation in PDA use treatment/recovery. Participants were recruited from different drug dependence treatment settings (residential and/or outpatient). Alcohol was the most reported substance in most of the articles included in this review, followed by opioids; the study participants were smokers. Treatment programmes where all offered freely in different state-owned or state-approved PDA use treatment facilities (see Table 1 for study details).

### Study quality assessment

The results of the quality assessment of qualitative studies using the 14-item NICE checklist show five studies [50, 51, 53, 54, 59] met all or most of the checklist criteria and were graded (++). Where they were not met, the reasons were either due to poor presentation of findings or lack of adequate details in their conclusions. Three studies [52, 55, 57] fulfilled some of the checklist criteria (8–10) and where they were not fulfilled or were inadequately described by the authors, the conclusions were unlikely to change had alternative approaches been adopted. These studies were graded (+). Two studies [58, 56] met few (six) of the checklist criteria, raising question whether the conclusions would likely or very likely change when alternative methodological and analytical approaches were used to conduct both studies. Both studies were graded (-).

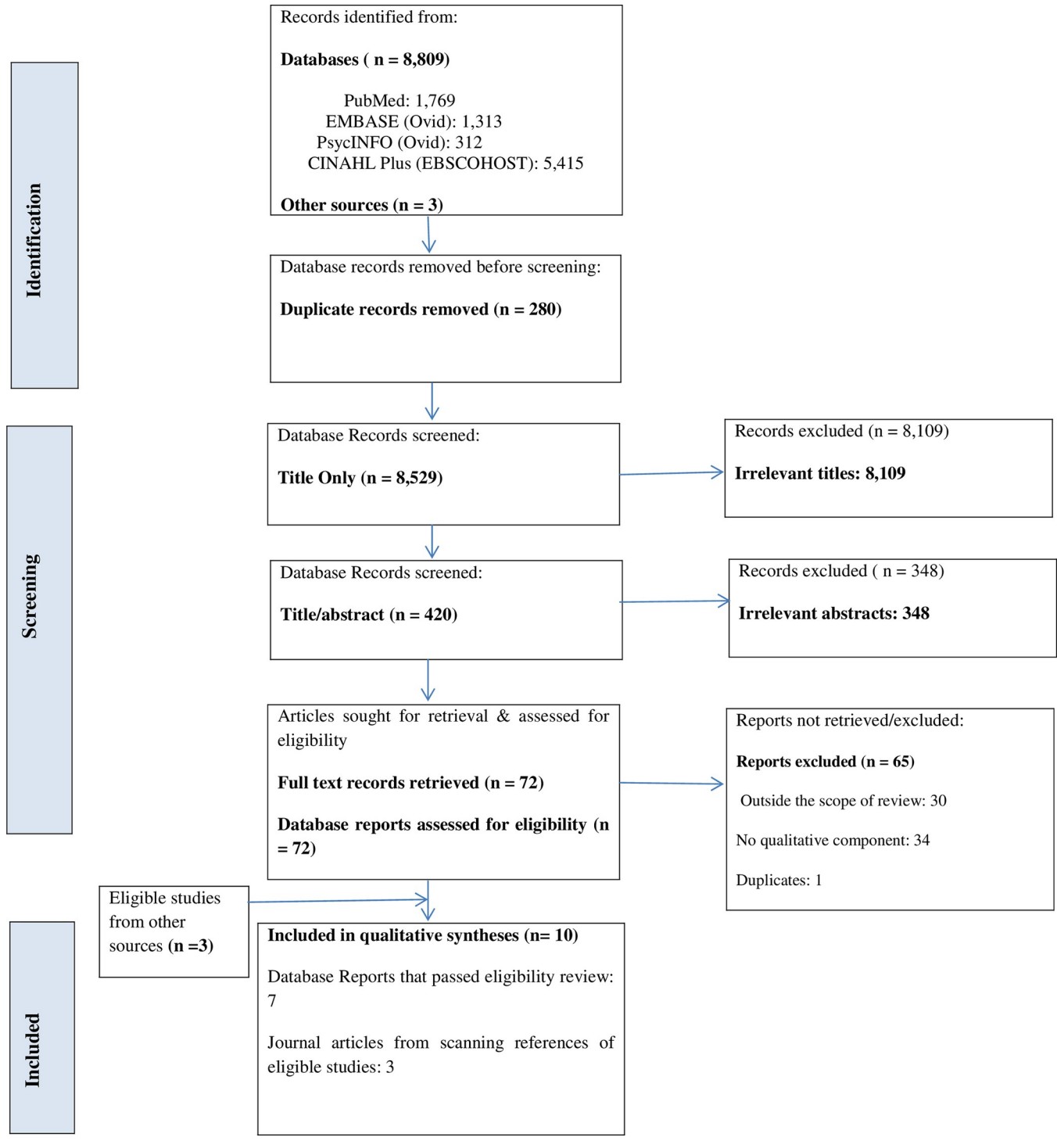

**Fig 1. PRISMA flow diagram [60].**

## Thematic synthesis

Thematic analysis of extracted smoking cessation related qualitative data identified three broad themes describing the barriers and two broad themes which described the facilitators.

**Table 1. Study details.**

| Lead author, publication year | Aim/focus of study, study design and analysis | Country | Sample | Intervention type and setting | Summary of findings and conclusions | Overall quality rating (based on NICE criteria) [46] |
|---|---|---|---|---|---|---|
| Berman et al. 2019 [50] | **Aim**: To (1) capture the views of people in community-based PDA use treatment programmes who use tobacco (2) identify how to tailor and plan both delayed and concurrent programmes to enhance smoking cessation **Study design**: Focus groups and individual interviews **Analysis**: Content analysis involving the use of pre-defined guide developed prior to data analysis based on scientific literature and stakeholders' experience of working with PDA use treatment community and involvement in group dialogues. | USA | 70.8% aged 39 years or younger 75% female 83.3% African American, 10.4% Hispanic/Latino 72.1% current smokers 16 in-recovery (former PDA use treatment) clients and 33 people in treatment for problematic drug or alcohol use. | 49 clients were attending community-based treatment programmes for PDA use | Almost all service users and providers expressed the common view about the inadequacy of stop smoking services in PDA use treatment settings, available to African Americans and other ethnic minority groups. It was recommended that stronger evidence of the benefits of the services to clients be put forward so that in general, stakeholders can possibly optimize availability, access, uptake/successful engagement with existing evidence-based smoking cessation services for all, and essentially tailor community cessation programmes to meet the needs of African American groups in PDA use treatment. | + + |
| Bhuiyan et al. 2017 [51] | **Aim**: To (1) identify the factors that motivate and hinder smoking cessation for people in treatment for PDA use (2) explore the attitude of people with PDA use towards social support while quitting smoking (3) identify where the social support offered during smoking cessation comes from (4) give a description of smoking cessation behaviours at the treatment setting **Study design**: Semi-structured, one-on-one interviews **Analysis**: Thematic analysis using grounded theory | USA | Mean age 44.2 (range 28–56) Male only 68.8% white, 25% Black/African American 75% use alcohol primarily 93.8% current smokers 43.8% smokers enrolled in cessation service 50% smokers not enrolled in cessation programme | NRT and one-on-one counselling in residential alcohol and other drug treatment settings | There were highlights of barriers to smoking cessation in PDA use treatment which were consistent with findings in the literature. They included treatment environment, programme implementation strategies and factors which affect confidence in quit success. It was suggested that programme administrators consider the adoption of policies which engender changes to culture and thus, optimize smoking cessation. This was said to have the potential of aiding the targeting of stop smoking services to population needs. | + + |
| Fallin et al. 2016 [52] | **Aim**: To give a description of (1) what prevents and encourage pregnant women who are receiving Medication Assisted Treatment (MAT) for opioid dependence to participate in tobacco treatment (2) the approaches to adapt tobacco cessation programmes to the needs of this group of people **Study design**: Semi-structured focus groups **Analysis**: Content analysis in MAXQDA using codebook developed before commencement of data analysis based on the experiences of the research team in the focus groups and literature | USA | Mean age 28 years *(range 22–37 years)* Female only All Caucasian 11 weeks to postpartum 86.4% current smokers 13.6% lifelong non-smokers | 22 participants enrolled in MAT (standard of care for opioid dependence) | Participants reported the desire and motivation to stop smoking but faced many barriers such as dependence on nicotine, use of tobacco to cope with stress and quitting the problematic use of more than one substance at the same time. The notable motivations to stop smoking included minimizing harm to others (strong family ties), unpleasantness of tobacco smoke and major life decision. Authors suggested that smoking cessation treatment be integrated into the psychosocial support offered alongside MAT for the client group, so that the provision of targeted treatment in a supportive environment can be made possible. | + |

*(Continued)*

**Table 1.** (Continued)

| Lead author, publication year | Aim/focus of study, study design and analysis | Country | Sample | Intervention type and setting | Summary of findings and conclusions | Overall quality rating (based on NICE criteria) [46] |
|---|---|---|---|---|---|---|
| Garner and Ratschen (2013) [58] | **Aim**: To explore the behaviours associated with smoking and quitting smoking, as well as the knowledge and experiences of homeless smokers who concurrently use other substances<br>**Study design**: Face-to-face interviews<br>**Analysis**: Framework analysis and descriptive statistics for narrative and structured data, respectively | UK | Mean age 33 *(range 18–53 years)*<br>73% male<br>27% female<br>53.3% alcohol users<br>40% used methadone<br>20% Amphetamine users | 15 homeless smokers with PDA use accessing a community harm reduction service | The homeless and highly dependent smokers were reported to lack adequate awareness/knowledge of the harms associated with smoking and thus, were given to high-risk smoking behaviours. The participants were confident they could stop smoking but in the past, had rarely received the necessary support to stop smoking from providers of services for the homeless and from other health professionals, especially for those motivated to do so. The necessity of adopting the practice of encouraging the uptake of tobacco harm reduction services/activities which could help tobacco use reduce/cut down smoking, was emphasized. | - |
| Kathuria et al. 2019 [53] | *Qualitative Component Only*<br>**Aim**: To<br>(1) explore existing practices associated with tobacco treatment during the inpatient Addiction Consult Service (ACS) visits<br>(2) identify/examine the barriers and facilitators to the delivery of smoking cessation support to hospitalized smokers with PDA use<br>(3) seek ways of improving stop smoking services during PDA use treatment interventions<br>**Study design**: Semi-structured interview with ACS physicians and PDA treatment patients<br>**Analysis**: Inductive content analysis with transcripts coded to consensus until thematic saturation | USA | *Patients*<br>55% male<br>35% were 44 years or younger<br>70% white, 15% Black/African America, 15% others<br>Only alcohol, 25% Alcohol and other drugs, 10%<br>*Physicians*<br>56% male<br>33% were 44 years or younger<br>89% whites (11% Hispanic/Latino), 11% Black/African American<br>56% completed clinical training 20 + years | 75.5% current smokers (≥ 1 cigarette(s) daily, one month prior) among hospitalized PDA use treatment patients, received ACS consultation (involving the delivery of NRT to 20.9% only) while hospitalized | During hospitalization, not much discussion about tobacco dependence took place, as reported by the patients and physicians. This was because physicians did not see tobacco use as an immediate public health threat. It was reported patients did not consider smoking cessation a priority, besides the constraints of time. Having a smoking-related health issue was considered a motivation for physicians to seek to address tobacco dependence. Suggestions were made and strategies were devised by the patients and physicians on how to integrate tobacco cessation programmes into treatment for PDA use. | + + |
| McCool and Richter (2003) [54] | **Aim**: To explore the diversity of views about the link between methadone, other drug use and smoking, to better understand why smoking is prevalent among people in treatment for drug dependence<br>**Study design**: Focus groups and individual interviews with different categories of smokers<br>**Analysis**: Pre-determined comments were entered into Microsoft Excel and underwent different levels of transcription, after which comments were categorized into four primary codes that showed the similarity or differences between smoking, alcohol, methadone and other drugs | USA | Average age, 44 years *(18 years or older)*<br>58% female<br>78% white<br>76% linked smoking to use of methadone, alcohol and other drugs<br>Average cigarettes/day, 25 *(SD 13)*<br>50% non-smokers for at least a day, in the past one year<br>Mean treatment duration in Methadone Maintenance Treatment (MMT), 11 years | Fifty-nine service users receiving Methadone Maintenance Treatment (MMT) continuously for at least the past two years | The majority of the patients associated smoking with methadone and other drug use. Patients viewed them as complementary and perceived they shared common onset cues/signals as well as withdrawal symptoms. They also attributed lesser consequences to the use of tobacco compared to other drugs. These views not only resulted in continued smoking but also drug dependence in a bi-directional sequence. The perceived interrelationship between the substances led to the suggestion that smoking cessation services should be encouraged in all PDA use treatment facilities and the use and effects of the substances addressed holistically. | + + |

*(Continued)*

**Table 1.** (Continued)

| Lead author, publication year | Aim/focus of study, study design and analysis | Country | Sample | Intervention type and setting | Summary of findings and conclusions | Overall quality rating (based on NICE criteria) [46] |
|---|---|---|---|---|---|---|
| Pagano et al. 2016 [55] | **Aim**: To examine the barriers to, and facilitators of tobacco control and tobacco cessation services as reported by stakeholders in a nationwide sample of drug dependence treatment interventions taking part in the 'NIDA Clinical Trials Network (CTN)' <br> **Study design**: Semi-structured interviews <br> **Analysis**: Thematic analysis based on grounded theory | USA | Mean age, 51years (SD 10.9) <br> 58.3% female <br> 83.3% White, 8.3% Native American, 4.2% Latino/Hispanic 4.2% African American <br> 69.5% postgraduate education <br> 17.4% current smokers | Participants in nine residential, seven methadone and eight outpatient treatment programmes 9.1% (n = 2) in recovery from PDA use | The barriers cited by the directors all of whom showed interest in helping clients stop smoking included clients' culture of smoking, resistance, unavailability of resources, smoking practices among treatment providers and environmental factors. They were also of the view that improved leadership, financial incentive and government action against smoking in drug dependence treatment programmes could support tobacco use cessation. Even though many PDA use treatment programmes are now placing more emphasis on smoking cessation, changes in infrastructure and culture are still needed, as observed. | + |
| Richter (2006) [56] | *Qualitative Component Only* <br> **Aim**: To present qualitative information about the attitude of treatment providers towards tobacco cessation initiatives in drug dependence treatment <br> **Study design**: Secondary analysis of a national survey involving the use of open-ended survey instruments <br> **Analysis**: The feedbacks were transcribed in Microsoft Excel and assessed for common themes, after which responses were grouped based on whether or not each common term was applicable to the treatment providers. An illustrative table was used to display quotes that exemplified each theme | USA | Involved the participation of 408 out of 697 outpatient MMT clinics: <br> 40% were for profit <br> 37% private/non-profit <br> 23% public clinic | Fourteen per cent of MMT clinic leaders believed patients benefitted from smoking in some ways | Many service providers were of the opinion that administering treatment for the other substance use problems was more important than smoking cessation. They also expressed the views that attempting smoking cessation by patients might not be appropriate under certain conditions such as when going through stress, in new treatment, reducing methadone dose or undergoing a detox. To this end, patients were advised not to attempt too many changes all at once. Some treatment providers thought tobacco use had some benefits when the goals were to manage some of the negative consequences of, and as an alternative to other drug use. | - |
| Richter et al. 2002 [57] | **Aim**: To <br> (1) identify how nicotine dependence treatment can be tailored to patients and <br> (2) examine the attractiveness and potential harm-reducing effects of smoking reduction and nicotine maintenance among people enrolled in MMT <br> **Study design**: Focus groups and individual interviews with different categories of smokers <br> **Analysis**: Focus group discussions and interviews were audio-recorded, pseudonyms assigned to speakers and text coded using a qualitative data analysis software, QSR Nudist IV | USA | Average age, 43.6 years <br> 57.7% female <br> 78.2% White <br> Average cigarettes/ day, 24.9 <br> 50% quit for at least a day in past year <br> Average years in MMT, 10.8 years | Seventy-eight service users in five MMT clinics | In general, clients who successfully stopped smoking used a combination of methods. Nicotine craving was the major trigger of tobacco re-uptake. Only few patients who had major health issues found NRT appealing/ useful, as they perceived they were unlikely to succeed at quit attempt. Some expressed fear that quitting could taper off MMT or distract them from addressing their drug dependence but others felt acquired skills could be applied to smoking cessation. The participants showed interest in reducing smoking as an alternative to cessation but those who had had failed attempts at reducing smoking preferred complete cessation. | + |

*(Continued)*

**Table 1.** (Continued)

| Lead author, publication year | Aim/focus of study, study design and analysis | Country | Sample | Intervention type and setting | Summary of findings and conclusions | Overall quality rating (based on NICE criteria) [46] |
|---|---|---|---|---|---|---|
| Wilson et al. 2015 [59] | **Aim**: To explore what clients and staff in drug dependence treatment setting believe about addressing tobacco dependence in PDA use treatment settings<br>**Study design**: Semi-structured interview with staff and focus groups with clients<br>**Analysis**: Constant comparative analysis. Transcripts were coded using NVivo 10 software (an inductive method) | Australia | *Treatment providers*<br>10 treatment providers from four programmes<br>Average age, 52 years *(range 32–65)*<br>60% female<br>20% current smokers<br>Average work duration, 15 years (range 1–32 years)<br>*Clients*<br>11 service users<br>91% male<br>All current smokers<br>Average age, 34 years *(range 24–53)*<br>Average cigarette/day, 14 *(range 7–30)* | Clients were enrolled in heroin, amphetamine or cannabis treatment programme | Treatment providers stated that organizational culture, lack of time and enforcement of smoking restrictions, the belief that tobacco smoking helps users cope with drug dependence treatment and that cessation is not a treatment priority for clients, as well as views it is both ineffective and under-utilized, are fundamental barriers to addressing tobacco dependence in PDA use treatment settings. Whereas clients associated smoking with habit, pleasure, seeing staff smoke, use of tobacco to relieve stress and concerns about NRT dependence besides its cost, as barriers. Both advocated for more smoking cessation efforts in PDA use treatment. | + + |

The barriers were perception of public health importance, programme structure, and intervention elements. The facilitators included supportive intervention environment as well as optimizing support/treatment staff resources.

## Perceived barriers to smoking cessation in PDA use treatment or recovery

Verbatim quotes representing participants' perception of the barriers people with PDA use face when seeking help to cut down or stop smoking in different PDA use treatment intervention settings are presented under each sub-theme.

**Perception of public health importance.** Two sub-themes perceived to influence people's perception of the public health importance and the consequent low smoking cessation service uptake and successful engagement were lack of priority focus and adequate communication.

*Priority issue*. A common perception from participants was the view that quitting tobacco use was not as important as getting off the substance which people were in treatment for, or recovering from. This perspective was shared by both PDA use treatment agencies/providers and services users. Many treatment programme administrators expressed concern that tobacco cessation could distract patients from getting off illicit drug use [57], while some thought it was more difficult to achieve [54]. In general, participants attributed less serious health consequences to tobacco use compared to their PDA use. This was commonly cited in residential treatment programmes [51, 53, 55].

> *"Alcohol's really toxic. If they come in with hepatitis, it's hard to argue we should be concentrating on smoking because they'll smoke for another 20 years. They're not going to live through this hospitalization if they don't stop drinking."* [53, p. 341] (Inpatient treatment provider quote)

In other contexts such as community drug/alcohol harm reduction programmes, service users widely reported that smoking cessation was not a priority to treatment providers or agencies [50, 58]. Some treatment providers simply advised patients to delay/defer quitting smoking for various reasons such as the additional responsibility on them to administer smoking cessation services, and fears that it could cause PDA use treatment/recovery relapse. Essentially, the main reason centered on smoking cessation not having the same priority as their PDA use [56, 59].

*". . . she says to me,' you're addressing other things at this time, I don't think you're ready to sort this out yet' (. . .), she made me not interested."* [58, p. 6] (Community harm reduction service user quote)

*Inadequate communication.* Smoking status was not considered a part of the initial patient assessment in many residential or inpatient PDA use treatment programmes because low priority was usually attributed to it [53, 59].

*"When we assess people first on a consult, we try to think of the whole plan right then. It would make sense to add smoking later, in a follow-up. We're not focused on it initially because of priority."* [53, p. 341] (Inpatient treatment provider quote)

In contrast, many treatment providers reported that discussions about smoking cessation occurred more frequently in outpatient PDA use treatment facilities even at first contact [59].

*"So they do get asked (about smoking) at assessment and we do discuss various techniques, the usual, you know set a date, goal setting, talk about getting NRTs."* [59, p. 95] (Outpatient treatment provider quote)

In other contexts such as community settings, information about smoking cessation was not conveyed in a manner that service users could understand the importance of stopping smoking or make them want to engage with smoking cessation services [50]. Some service users recommended that awareness about the health consequences of tobacco consumption can be better enhanced by using graphics or visual displays of damages to major organs such as the lungs. Other participants accessing similar community services advocated for testimonials or visits to hospitals where people are sick or dying from tobacco-related health issues [50].

**Programme structure.** Time and environmental factors inherent in the design and delivery of smoking cessation services in PDA use treatment programmes were commonly cited as barriers to uptake and successful engagement.

*Timing.* Clients and staff of different PDA use treatment programmes perceived the timing was not right to utilize or engage with existing stop smoking services at the same time people were trying to address their PDA use [50, 52, 57]. Some residential treatment providers expressed the view that attempting to address all problematic substance use at the same time can be overwhelming for service users [53, 54, 56]. Other treatment providers also shared concerns that smoking cessation in the early stages of recovery from PDA use can be stressful and overwhelming for clients [52]. These views were corroborated by patients enrolled in residential and outpatient treatment programmes, respectively [52, 57].

*". . . 'There are more important issues at hand', was what he [staff] was saying. 'Just don't use drugs. . . just don't worry about the cigarettes right now, because there are bigger fish to fry, so to speak'."* [57, p. s178] (Outpatient service user quote)

*Environmental influence*. Participants from all of the included studies expressed different views about the impact of the environment where people were receiving support for their PDA use on their ability to successfully engage with stop smoking support. Some respondents were of the opinion that mandatory smoking bans/restrictions in treatment facilities encouraged uptake of stop smoking services. Other participants expressed the view that the smoking restrictions pose a greater challenge for people who prefer to gradually cut-down smoking rather than outright cessation [50, 51, 53].

*"If I come off a bus into a programme and nobody's smoking... no, I'm not gonna want to smoke."* [50, p. 8] (Community service user quote)

Another view expressed was that smoking was a culturally acceptable norm in the 'recovery community', involving both treatment providers and service users. They likened it to a normal activity that has gained wide acceptance as a form of drug use in recovery programmes. This is because both treatment providers and service users usually consumed tobacco in settings where people were in recovery from PDA use [51, 54]. Most directors of PDA use residential treatment programmes corroborated this view [55] but felt hypocritical about it [59]. Many treatment providers reported the culture of staff smoking whereas they offered advice to service users against smoking while in treatment/recovery from PDA use. This action discouraged many service users who wanted to stop smoking [55, 59].

*"...In certain recovery settings the staff also will take a smoke break with the population and stuff like that ... literally everybody in the building ... it's just like a norm ... it just seems like it's accepted."* [51, p. 1086] (Residential service user quote)

**Intervention elements.** Services providers and users of PDA use treatment services reported different outcomes of some smoking cessation models such as the use of NRT and quit lines. The majority of the clients were of the view that the smoking cessation methods were not very effective. Whereas, most treatment providers described how limited time and resources affected availability and access to smoking cessation services, as well as how it affected clients' uptake or successful engagement behavior [50, 52, 57, 59].

*Effectiveness of smoking cessation models*. Participants from different studies reported different outcomes of the various smoking cessation methods. Many expressed the view that NRT and other pharmaceuticals such as Varenicline (Champix) and quit lines were not effective. Some service providers and users complained that the nicotine patches or gum did not help them stop smoking [50, 52, 57, 59]. Some commonly cited reasons for service users' perception of the ineffectiveness of smoking cessation approaches included limited cessation service options, poor explanation of quit line services and benefits. Others were the experience or perception of side effects from the use of pharmacotherapy such as NRT and Varenicline. [50, 52, 57, 59].

*"Well my (own quitting) experience of NRT has been pretty disappointing. I didn't really find that it helped at all. But then again I've heard other people say that it has helped but, no, I don't think it was fantastic at all. I wouldn't use that method again."* [59, p. 95] (Community-based treatment provider quote)

Some service providers in a community drug harm reduction programme expressed some views which they perceived were the reasons for the negative experiences of service users with the different smoking cessation methods. The reasons included service users' lack of

experience with or access to comprehensive smoking cessation services which combine NRT and in-depth counselling and also promote sustained quit lines [50]. In other contexts such as residential PDA use treatment settings, service providers and users expressed the view that behavioural counselling was more effective when patients are stable from the effects of withdrawing the illicit substance, following the period of hospitalization [53].

> *"Give [me] a couple of days. I got mad when they first mentioned cigarettes.. . .I told them I wasn't ready, but now [a few days later] I'm ready."* [53, p. 343] (Inpatient service user quote)

*Intensity and coverage of smoking cessation services.* Smoking cessation opportunities and approaches were reported to be rarely discussed with service users when they are newly enrolled in PDA use treatment/recovery programmes. Where such conversations were held, they were either not engaging enough or the smoking cessation services available did not include methods/products that patients could have preferentially used and possibly engage with. The most commonly cited reason for the preferences was clients' report of negative experiences with the use of some smoking cessation methods/products in the past [50, 53–55]. Residential treatment providers also cited lack of resources for staff training and implementation of counseling groups as key barriers that made it difficult to deliver smoking cessation services or willingly offer support when it was their mandate to do so [55, 59].

> *"So we have the culture from the staff, how they were trained in schools, on the one hand, and we have the culture of the clientele, both of those work against smoking cessation."* [55, p. 24] (MMT provider quote)

It was commonly reported by both health care providers and service users that the costs of the different smoking cessation products such as nicotine patches or gums were not fully covered by the comprehensive smoking cessation services in different PDA use treatment settings. The tobacco cessation treatment options sometimes varied by service users' insurance plans which limited the smoking cessation services they could benefit from and engage with [55, 50]. In addition to this, the free smoking cessation support offered did not extend beyond the residential period for patients in residential/inpatient programmes [57] or to the follow-up period for those accessing outpatient services [54].

> *"'You're all on your own afterwards.' The hospital did not provide take-home patches or any form of after-care for nicotine dependence."* [57, p. S178] (Inpatient service user quote)

**Perceived facilitators.**   The provision of a supportive treatment environment and optimizing support/staff resources for smoking cessation service delivery were the commonly expressed views that could potentially enhance uptake of, and successful engagement with smoking cessation services while in treatment/recovery from PDA use. Verbatim quotes representing participants' perception of what facilitates smoking cessation service uptake in different PDA use treatment intervention settings are presented under each sub-theme.

**Supportive intervention environment.**   The participants expressed mixed views about the impact of smoking restrictions during PDA use treatment/recovery (theme 2) but agreed on the need for smoking cessation support to be properly integrated into drug dependence treatment by prioritizing it [53]. Some participants of outpatient programmes also expressed the views that designing smoking cessation services in ways that service users can easily engage with them could facilitate uptake [51, 52, 55, 57]. This could be achieved by having external

support from people (like the sponsors in drug dependence treatment programmes). They can closely monitor service users' progress and readily provide motivations to continue with their efforts at stopping smoking while in treatment/recovery from PDA use [52].

*Prioritizing smoking cessation support.* Treatment providers expressed some views about actions they perceived could help prioritize smoking cessation support in PDA use interventions and thus, facilitate uptake. Some of the recommended actions included frequent communication, defining drug dependence to include tobacco consumption, and integrating smoking cessation support into smoking-related health issues (essentially for clinic-based interventions) [53, 54].

> *"I think when you separate out tobacco from other substances; it really makes it a different— it's like a more benign kind of thing when you and I know that it's not."* [53, p. 342] (Inpatient treatment provider quote)

*Tailoring stop smoking services.* Whereas some service users stated that they felt motivated to stop smoking completely by reducing the number of cigarettes they smoked daily, others perceived smoking reduction was ineffective or difficult to sustain and therefore, preferred cessation [57]. Many service users were also of the opinion that increasing the frequency of smoking cessation service delivery (to say, more than once a week) as well as adopting a group-like smoking counseling sessions could facilitate the rate of successful engagement [51].

> *"We know, up here, that we probably can't smoke just two or three cigarettes a day. It would be ideal if we could, but we probably can't. But we'd sure like it if we could. But I know I couldn't, ever.... It's not realistic for me. Damn it, I've tried*! *It would have to be all or nothing at all."* [57, p. S180] (MMT service user quote)

Health care providers and users of different PDA use treatment clinics unanimously advocated for more intensive tobacco cessation support. They recommended the creation of specialist units within PDA use treatment programmes or agencies. This should comprise people who can provide 'real-time' support [52], act as role models or mentors (being successful quitters), and can effectively communicate the contents of a smoking cessation message [51, 55]. These specialist units were also expected to help maintain engagement through a 'sponsorship role' which is similar to what obtains in other drug dependence treatment programmes [52].

> *"[I wish there was] a person you could talk to more about smoking. A person that knows about quitting and gives out some advice to stop smoking"* [51, p. 1086] (Residential service user quote)

**Optimizing support/treatment staff resources.** Integrated care through increased staff support was reported to have helped prioritize smoking cessation services for people in PDA use treatment/recovery by surmounting the barriers of time and resource constraints. It also facilitated the delivery of smoking cessation services using the best approaches because of enhanced communication activities [53, 57].

*Team-based strategy.* Many residential programme administrators said they believed that a team-based approach reduces the burden on treatment providers to deliver smoking cessation support alongside treatment for PDA use [53, 59]. Some service providers acknowledged that many of the general staff might have skills that can be deployed to deliver smoking cessation [53].

> *"I think everybody on the care team should support smoking cessation. I think for the patients for whom we're already having in depth discussions about their substance use, the tobacco*

*should be part of it. I also think they should not wait for the consult service for medication, NRT to start."* [53, p. 342] (Inpatient treatment provider quote)

Participants perceived it would be much easier to determine the best timeline to deliver smoking cessation services through this approach. For example, clients and treatment providers in residential programmes reported positive outcomes when discussions about smoking cessation commenced when patients have been stabilized (from substance withdrawal symptoms) [53].

*"Give [me] a couple of days. I got mad when they first mentioned cigarettes.. . .I told them I wasn't ready, but now [a few days later] I'm ready."* [53, p. 343] (Inpatient service user quote)

In general, many programme directors reported that financial support for delivering stop smoking services including acquiring medications, sustaining engagement (follow-up inclusive), helped increase service users' uptake and successful engagement [55]. The use of incentives and a reward-based system were perceived as great facilitators of smoking cessation service uptake for people in PDA use treatment/recovery. The reward system included awards of certificates, creation of unique free-time activities for people who had partaken in smoking cessation sessions or have stopped smoking for a period of at least thirty days [55].

*Positive communication.* Stakeholders described the importance of using appropriate language to communicate smoking cessation activities/plans to service users. Some treatment providers proposed that the message should be framed in ways that conveyed a positive outlook capable of inspiring treatment clients' confidence in the support offered and their capacity to engage successfully [53, 57]. Many treatment providers noted that framing the message along the line of thought that 'patients who quit tobacco use would be more successful in stopping the use of other drugs or alcohol', might motivate them to quit all the substances including tobacco [53]. This view was also expressed by patients receiving methadone treatment for opioid dependence [57].

*"There are many observational studies that show that the people who quit smoking do better.' That's what I explain to patients and actually patients are very open to hearing that."* [53, p. 342] (Inpatient treatment provider quote)

Congruently, a patient who was receiving methadone treatment for opioid dependence reported that the encouraging tone used by a methadone-dispensing nurse helped him successfully engage with tobacco cessation services and stay quit, upon notifying her of his intention to stop smoking [57].

*"She just asked, 'How?' I told her that I was thinking about getting on the patch and she said she thought it was fine. She would ask me occasionally how I was doing and if I was still on the patch or whatever. Occasionally, I would tell her that it has been a month, three months; it has been a year or whatever. She would just tell me how proud she was of me."* [57, p. S178] (MMT service user quote)

## Discussion

This is the first qualitative systematic review to examine the barriers and facilitators to uptake and engagement with existing stop smoking services for people in treatment/recovery from problematic drug or alcohol use. Results from the thematic analysis indicated that treatment

providers and users' perception of the public health importance of smoking cessation, programme structure, and intervention elements were the commonly reported barriers. Supportive intervention environment and optimization of support/staff resources for smoking cessation services represented key facilitators of patients' successful engagement with stop smoking services. These broad themes had complex interacting sub-themes which altogether underscored the need to change narratives, create the right environment and consider a comprehensive care approach to addressing the problem. The results of this study will not only provide the contextual details that were lacking in other systematic reviews, but will also help inform future intervention development to help this group cut down or stop smoking in the future.

## Changing the narratives

Qualitative evidence from this systematic review has shown that only few of the people in PDA use treatment/recovery programmes actually know the importance of smoking cessation, as opposed to popular views that they are not always willing to consider it [61, 62]. This is because the attention given to smoking cessation and support is limited. It is less and poorly talked about, undermining its relevance to public health. Empirical evidence from observational studies support these views [63, 64].

The evidence that communicating health service strategies in a manner that is both engaging and gives people a better sense of awareness of how relevant the health issue and their responses are to public health, is well reported in the literature [65, 66]. This strategy could increase the chances for successful engagement by taking the harm reduction intention (cessation or reduction) of people in PDA use treatment/recovery into consideration [61].

## Creating suitable environment

Environmental factors such as the culture of smoking and delivering smoking cessation support within PDA use treatment were reported to have limited clients' access to, or engagement with stop smoking services [50, 51, 53, 55]. This is considered a major limiting factor associated with PDA use treatment implementation [67, 68]. The mixed views about environmental factors have also been extensively examined in systematic reviews [69, 70] and different quantitative studies [1, 11, 62, 71, 72].

The impact of these 'environmental' factors on PDA use treatment/recovery outcomes is well recognized [73, 74]. People's social networks exert more influence on their behaviour the closer these networks are to them [75]. The specialist units many respondents in our qualitative synthesis advocated for which they perceived comprise 'champions' [51, 55], could exert this influence. Bringing in an external stop smoking advisor/service and/or peer supporters such as clients who have stopped smoking might be needful in delivering smoking cessation services within PDA use treatment programme structure.

## Considering comprehensive care

There is a considerable existing evidence base demonstrating the effectiveness of pharmacotherapy in smoking cessation, either alone or in combination with behavioural approaches [4–6]. Availability and accessibility of smoking cessation approaches differed based on the free smoking cessation services that were comprehensively offered [50, 55, 57]. This makes it difficult to achieve smoking cessation goals within PDA use treatment programme structure because better outcomes in drug dependence treatment are derived for example, from medium to long-term support [76, 77]. Currently, no evidence explains what the scope of stop smoking

services (quality and access) implies for people with PDA use that are in recovery or have left the residential treatment facility.

In general, the way PDA use treatment services are offered does not prioritize smoking cessation services over the other treatment demands it competes for [63, 64, 78]. Studies have recommended a comprehensive care approach which requires detailed assessment of needs and a flexible recovery plan that can be reviewed [77, 79–81]. This approach allows tailoring stop smoking services to meet clients' needs [82] especially in resource-constrained settings [83]. Evidence supports the idea that integrated care can significantly influence smoking cessation/reduction outcomes because it offers support along the continuum of care [84–86]. This is needed to improve treatment quality and clients' access/engagement [81].

## Strengths and limitations

A particular strength of this systematic review is its focus on qualitative studies, with existing quantitative studies mostly focused on outcome evaluation of smoking cessation services. Evidence suggests that people disengage from treatment when they do not feel motivated to continue, or feel unsatisfied with treatment or when the support is not sustained [67, 87]. Future qualitative studies should seek to examine in greater detail, perspectives from community-based and outpatient services.

A significant limitation of this systematic review is that it does not include as much qualitative evidence from community-based and outpatient contexts as residential/inpatient harm reduction services. Studies examining the effectiveness of smoking cessation interventions in these contexts are limited, despite a significant reduction in the number of residential visits as observed in the USA and UK for example [36, 88–90]. This represents a significant research gap in the literature that future researches need to address.

Another limitation is that this review has not synthesized evidence from low-income countries. Generally, studies that have examined stop smoking behaviours in the relevant cultural and socioeconomic contexts of developing countries where the burden of smoking is highest [12] are limited [91]. This gap in literature affects the transferability of findings from this systematic review to low-income settings.

## Conclusion

Public health practitioners/commissioners working in smoking cessation services need to examine evidence from all drug dependence treatment contexts. This will help them to adequately devise and assess the feasibility of targeted tobacco harm reduction strategies which are aimed at assisting a stigmatized population to reduce or stop smoking. PDA use treatment providers need to influence a change in the way people perceive the importance of smoking cessation activities during PDA use treatment/recovery. They also need to place more emphasis on the need to create suitable environment for sustained adherence to treatment/recovery plans as well as deliver PDA use treatment interventions within the health system as comprehensive care. This will enhance patients' uptake and successful engagement with existing stop smoking services especially in residential/inpatient treatment programmes. These efforts would ease the burden on health care systems from the health issues and preventable deaths that are direct consequences of tobacco use.

## Supporting information

**S1 Appendix. Study (systematic review) protocol.**
(PDF)

**S2 Appendix. NICE checklist guidelines.**
(DOCX)

**S1 Fig. Databases search strategy/terms.**
(TIF)

**S1 File. PRISMA 2020 checklist.**
(DOCX)

**S2 File. ENTREQ checklist.**
(DOCX)

**S3 File. NICE checklist.**
(DOCX)

## Author Contributions

**Conceptualization:** Eric Osayemwenre Iyahen, Neneh Rowa-Dewar, Fiona Dobbie.

**Data curation:** Eric Osayemwenre Iyahen, Osazee Omoregbe Omoruyi.

**Formal analysis:** Eric Osayemwenre Iyahen, Osazee Omoregbe Omoruyi, Fiona Dobbie.

**Methodology:** Eric Osayemwenre Iyahen, Osazee Omoregbe Omoruyi, Neneh Rowa-Dewar, Fiona Dobbie.

**Supervision:** Neneh Rowa-Dewar, Fiona Dobbie.

**Writing – original draft:** Eric Osayemwenre Iyahen.

**Writing – review & editing:** Eric Osayemwenre Iyahen, Osazee Omoregbe Omoruyi, Neneh Rowa-Dewar, Fiona Dobbie.

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
