## [Decision Letter · Decision Letter 0]

3 May 2023

PONE-D-23-04818Exploring the barriers and facilitators to the uptake of smoking cessation services for people in treatment or recovery from problematic drug or alcohol use: A qualitative systematic reviewPLOS ONE

Dear Dr. Iyahen,

Thank you for submitting your manuscript to PLOS ONE. After careful consideration, we feel that it has merit but does not fully meet PLOS ONE’s publication criteria as it currently stands. Therefore, we invite you to submit a revised version of the manuscript that addresses the points raised during the review process. The manuscript has been evaluated by one reviewer, and their comments are available below.

The reviewer has raised concerns regarding the reporting and language of the study. 

Could you please revise the manuscript to carefully address the concerns raised?

Please note that we have only been able to secure a single reviewer to assess your manuscript. We are issuing a decision on your manuscript at this point to prevent further delays in the evaluation of your manuscript. Please be aware that the editor who handles your revised manuscript might find it necessary to invite additional reviewers to assess this work once the revised manuscript is submitted. However, we will aim to proceed on the basis of this single review if possible.

We look forward to receiving your revised manuscript.

Kind regards,

Johannes Stortz

Staff Editor

PLOS ONE

Reviewers' comments:

Reviewer's Responses to Questions

**Comments to the Author**

1. Is the manuscript technically sound, and do the data support the conclusions?

Reviewer #1: Yes

2. Has the statistical analysis been performed appropriately and rigorously? 

Reviewer #1: N/A

3. Have the authors made all data underlying the findings in their manuscript fully available?

Reviewer #1: Yes

4. Is the manuscript presented in an intelligible fashion and written in standard English?

Reviewer #1: Yes

5. Review Comments to the Author

Reviewer #1: Thank you for the opportunity to review this manuscript.

This systematic review addresses the factors that hinder or enhance the delivery of smoking cessation treatment to people with problem drinking or drug use behavior. It is an important topic with significant public health implications given that smoking remains the leading cause of preventable death even among those with problem alcohol or drug use. The systematic review follows appropriate guidelines and methodologies and the conclusions follow logically from the data extracted from the 10 papers that were reviewed.

My have only 3 minor suggestions to improve the manuscript:

1) The paper would greatly benefit from grammatical review. There is a dizzying number of commas and misplaced colons and semicolons that result in poor sentence structure throughout the paper. There are also a number of typos that are distracting (e.g., Page 9, Line 182, should refer to 420 database records rather than 420 databases, which would have been an extraordinary amount of work!).

2) The manuscript is quite long, even for a systematic review. I would suggest trimming the number of quotes presented in each section to only those that offer novel information in support of the theme. I would also encourage the authors to find ways to streamline the Discussion.

3) Page 2, line 21: This is the first place that the acronym PAD is introduced as referring to "problematic drug or alcohol use." This phrasing would suggest that the acronym should be PDA. Suggest rewording here and in other places where the acronym is defined.

6. PLOS authors have the option to publish the peer review history of their article (what does this mean?). If published, this will include your full peer review and any attached files.

Reviewer #1: No

---

## [Author Response · Author response to Decision Letter 0]

31 May 2023

1. We have carried out an extensive grammatical review of the paper. This included copy editing to reflect proper punctuations and improved sentence structure. Also, we have changed the text on Page 9, Line 182 to 420 database records.

2. We have trimmed the number of quotes in the results section to reflect only those that offer novel information in support of the themes. The discussion section has also been rephrased to reduce word count and the overall length of the manuscript.

3. Apologies for this oversight, this has been rectified. The acronym PAD has been replaced with PDA throughout the manuscript.

---

## [Editor Report · Decision Letter 1]

18 Jun 2023

PONE-D-23-04818R1Exploring the barriers and facilitators to the uptake of smoking cessation services for people in treatment or recovery from problematic drug or alcohol use: A qualitative systematic reviewPLOS ONE

Dear Dr. Iyahen,

Thank you for submitting your manuscript to PLOS ONE. After careful consideration, we feel that it has merit but does not fully meet PLOS ONE’s publication criteria as it currently stands. Therefore, we invite you to submit a revised version of the manuscript that addresses the points raised during the review process.

Many thanks for attempting to address the reviewers' comments.  As academic editor I have reviewed the revisions and I think there is still more work to be undertaken in terms of reducing the overall length of the manuscript and shortening the length of sentences.  Please try and limit the use of quotes to one example per point, rather than in many places there are currently two quotes.  The discussion has not been sufficiently reduced and I would like this to be consolidated further. 

We look forward to receiving your revised manuscript.

Kind regards,

Charlotte Lennox

Academic Editor

PLOS ONE
---

## [Author Response · Author response to Decision Letter 1]

23 Jun 2023

We have carried out another round of copy editing to reflect proper punctuations and improved sentence structure. The length of sentences in the manuscript has been shortened because of this.

We have limited the use of quotes in the results section to only one quote to buttress each point made. Also, the discussion section has been further trimmed down to reduce word count and the overall length of the manuscript.

---

## [Editor Report · Decision Letter 2]

26 Jun 2023

Exploring the barriers and facilitators to the uptake of smoking cessation services for people in treatment or recovery from problematic drug or alcohol use: A qualitative systematic review

PONE-D-23-04818R2

Dear Dr. Iyahen,

We’re pleased to inform you that your manuscript has been judged scientifically suitable for publication and will be formally accepted for publication once it meets all outstanding technical requirements.

Kind regards,

Charlotte Lennox

Academic Editor

PLOS ONE
---

## [Editor Report · Acceptance letter]

4 Jul 2023

PONE-D-23-04818R2 

Exploring the barriers and facilitators to the uptake of smoking cessation services for people in treatment or recovery from problematic drug or alcohol use: A qualitative systematic review 

Dear Dr. Iyahen:

I'm pleased to inform you that your manuscript has been deemed suitable for publication in PLOS ONE. Congratulations! Your manuscript is now with our production department. 

Kind regards, 

on behalf of

Dr. Charlotte Lennox 

Academic Editor

PLOS ONE